# Association between Menopausal Women’s Quality of Life and Aging Anxiety: The Role of Life Satisfaction and Depression

**DOI:** 10.3390/medicina60081189

**Published:** 2024-07-23

**Authors:** Seunghee Lee, Mijung Jang, Dohhee Kim, KyooSang Kim

**Affiliations:** 1Department of Research Institute, Seoul Medical Center, 156, Sinnae-ro, Jungnang-gu, Seoul 02053, Republic of Korea; shlee282@seoulmc.or.kr (S.L.); mijungjang@seoulmc.or.kr (M.J.); dohheekim@seoulmc.or.kr (D.K.); 2Department of Public Health, Graduate School, Korea University, 73 Goryeodae-ro, Seongbuk-gu, Seoul 02841, Republic of Korea; 3Department of Occupational Environmental Medicine, Seoul Medical Center, 156, Sinnae-ro, Jungnang-gu, Seoul 02053, Republic of Korea

**Keywords:** menopausal quality of life, aging anxiety, life satisfaction, depression, middle-aged women

## Abstract

*Background and Objectives*: This study investigated the links among quality of life, life satisfaction, depression, and aging anxiety in menopausal middle-aged women. The objective was to establish an understanding of how these factors are associated, which would be the foundation for developing programs aimed at enhancing the health and well-being of menopausal women. *Materials and Methods*: An online survey was administered to 993 middle-aged women, aged 45 to 65, residing in Seoul, Korea. The survey evaluated menopausal quality of life, life satisfaction, depression, and aging anxiety. Additionally, a Process Macro Model 4 was used to assess the links between life satisfaction, depression, menopausal quality of life, and aging anxiety. *Results*: Aging anxiety in middle-aged women was associated with a lower score on the menopausal quality of life scale (r = 0.37, *p* < 0.001), lower life satisfaction (r = −0.46, *p* < 0.001), and higher depression (r = 0.42, *p* < 0.001). In addition, there was an indirect effect—mediated by depression (95% CI = 0.025, 0.058) and life satisfaction (95% CI = 0.038, 0.064)—between menopausal quality of life and aging anxiety. *Conclusions*: The present study demonstrated a direct effect of low menopausal quality of life on aging anxiety and a mediating effect of low depression and higher life satisfaction on aging anxiety. These results suggest the need for programs to increase menopausal quality of life, decrease depression and improve life satisfaction to reduce aging anxiety.

## 1. Introduction

According to Statistics Korea’s “Statistics on the Elderly in 2022”, Korea will enter an ultra-elderly society in 2025.The pace of aging is extremely rapid [1]. Entry into an ultra-elderly society, coupled with a declining birth rate, could lead to problems such as a dwindling productive population, increasing social security costs, and slow economic growth which require urgent intervention measures at the national and individual levels. The growing interest in healthy and successful aging, as opposed to simply prolonging life, has led to increased research on the concept and its influencing factors. Rowe and Kahn proposed a model of successful aging that include (1) low risk factors for disease and disability, (2) maintenance of high physical and mental function, and (3) active participation in life. The definition of successful aging is when these three domain conditions are met [2].

Preparing for old age is important for successful aging, and most people experience anxiety when this preparation is lacking. The complex concept of worrying about getting older is called aging anxiety, and it is a situation which can become acute in midlife [3]. Midlife occurs before the onset of old age, a time of declining physical abilities and social and emotional changes about growing old and retirement. It has been reported that middle-aged adults (Age 25–65) are more negative and anxious about aging than older adults (Age 66–74 years) [4], and have the highest levels of aging anxiety of all age groups [5]. Although earlier studies on aging anxiety have targeted mainly older adults, recent research has shifted to examining the factors contributing to successful aging in middle-aged individuals [6,7]. Findings indicate that aging anxiety is significantly associated with how individuals adapt to aging. Anxiety and fear of aging are associated with negative life experiences, whereas positive attitudes promote successful aging [8].

In middle-aged women, menopause begins due to a decline in ovarian function; the lack of estrogen leads to health problems such as psychological instability [9]. The periods before and after menopause are called perimenopause and postmenopause, respectively. The symptoms of perimenopause include hot flashes, urinary incontinence, anxiety, joint pain, memory problems, and sleep disturbances. More than 50% of middle-aged women in Korea report symptoms of menopause. These symptoms have been reported to interfere with daily life, decrease quality of life [10,11], and increase aging anxiety [12]. While previous research has explored the link between menopausal symptoms and depression, there has been limited focus on the relationship between menopausal quality of life and aging anxiety. Further research is needed to examine the impact of quality of life on aging anxiety in menopausal women.

The concept of life satisfaction was first used by Neugarten [13], who defined it as experiencing joy in daily life, having a sense of meaning, responsibility, and purposeful fulfillment in one’s life, maintaining optimistic attitudes, having a positive self-image and seeing oneself as valuable. Life satisfaction is a broad subjective spectrum encompassing the present and the past, consistent with the concepts of happiness, quality of life, and successful aging, and is used and defined variously by scholars. Research on life satisfaction and aging anxiety has shown that life satisfaction affects death anxiety, and stress, depression, and other negative thoughts about life [14] are closely related to aging anxiety.

Depression in menopausal women has been attributed to hormonal changes, physical symptoms, changes in family roles, and alterations in appearance [15]. Previous research indicates that aging anxiety is significantly linked to high levels of depression, which adversely impact quality of life [16]. However, due to variations in racial and country-specific characteristics, there is a lack of studies examining the relationship between menopausal quality of life and aging anxiety among Korean women. It is essential to analyze the mediating effects not only of depression, but also of the positive factor of life satisfaction. Some studies have reported that the physical health of middle-aged women is a crucial factor influencing their mental health [17,18]. Physical health can be assessed through subjective health status reports, body mass index (BMI), and the presence or absence of chronic diseases. However, research findings on the impact of body mass on mental health vary. According to Lee and Ma [17], a high BMI has a negative impact on women’s quality of life and chronic diseases. Their study reported that increased levels of obesity correlate with higher levels of stress, depression, and anxiety. Conversely, Bang and Do’s study found that middle-aged women who were underweight had a lower quality of life [18].

Mediating effects can identify which variables strengthen or weaken the relationship between an independent variable and a dependent variable, helping to explain complex relationships in social sciences and psychology more clearly. Cao et al. [19] identified the significant mediating role of self-management in the relationship between quality of life and anxiety. In line with this finding, the present study seeks to explore how life satisfaction, a measure of subjective happiness and a positive outlook, and depression, which intensifies negative emotions, mediate these associations. These variables are crucial indicators of individuals’ perceptions of their lives and their ability to manage emotional challenges. By addressing these factors effectively, our aim is to enhance quality of life and alleviate anxiety. A direct effect is the path from an independent variable directly to a dependent variable without going through a mediator, while an indirect effect is a path through a mediator.

Previous studies have suggested an association between quality of life, depression, and aging anxiety. However, few studies have examined menopause-specific quality of life and aging anxiety among middle-aged Korean women, taking into account cultural and demographic factors and the mediating roles of life satisfaction and depression.

The findings will help confirm these relationships and provide a foundation for developing preventive programs designed to improve negative perceptions of aging and to promote health in later life.

## 2. Materials and Methods

### 2.1. Participants

This study utilized a web survey to collect data from middle-aged women aged 45 to 65, living in Seoul. The aim was to understand the impact of general characteristics, menopausal quality of life, life satisfaction, and depression on aging anxiety. Using G*Power 3.1.9.7 for multiple regression analysis, a sample size of 194 was required based on an effect size of 0.15, a significance level of 0.05, and a power of 0.95. Ultimately, 993 out of 1003 panel members, registered with the survey company, participated, with data from 10 unfaithful respondents excluded from the final analysis. 

### 2.2. Research Tools

#### 2.2.1. Aging Anxiety

The Anxiety about Aging Scale (AAS), originally developed by Lasher and Faulkender [20], consists of 20 items measured on a 5-point Likert scale. For this study, the scale was modified by Kim Wook [21] to a 19-item version by excluding one item due to low reliability. Permission for the adaptation was obtained via email. Negative items were reverse scaled and reflected as mean values, with higher scores indicating higher aging anxiety. There were four subscales: fear of older people, psychological instability, worry about appearance, and fear of loss, with a Cronbach’s α of 0.62 for reliability.

#### 2.2.2. Korean Version of the Menopause-Specific Quality of Life Assessment

The Menopause-Specific Quality of Life (MENQOL) measurement tool, which is used to quantify and analyze the quality of life of menopausal women, was developed by Hilditch et al. [22], revised by Lewis et al. [23] and has been translated into Korean. The original version had 29 questions, but in this study the Korean version of the Menopause-Specific Quality of Life (MENQOL-K) measurement tool—verified by Park Jin-hee [24]—was used. It has 28 questions—excluding the item “hair on the face”, which is not related to the quality of life of Korean women in menopause—dealing with vascular movement (3 questions), psychosocial aspects (7 questions), body aspects (15 questions), and sexual life aspects (3 questions). The purpose of, and necessity for, the study were explained through e-mail and approved for use. The presence or absence of symptoms was measured. If there were no symptoms, a score of 1 point was given. If symptoms were present, they were rated on a 7-point scale from 2 (mild) to 8 (severe). A higher score indicates more severe symptoms and thus a poorer quality of life related to menopause. In this study, Cronbach’s α value was 0.91.

#### 2.2.3. Life Satisfaction

Diener’s [25] Satisfaction with Life Scale (SWLS) is a tool that consists of five questions that can simply and quickly evaluate the life satisfaction of respondents. It is a 5-point Likert scale measuring from “not at all” (1 point) to “very much” (5 points) according to the degree of feeling; a higher score implies higher life satisfaction and Cronbach’s α value was 0.91.

#### 2.2.4. Depression

The Center for Epidemiological Studies Depression Scale (CESD-11), developed by Kohout [26], is a tool to evaluate depression in the public. CESD-11 consists of 11 items. The 11 response measures were calculated with a score from “extremely rare” (1 point) to “mostly so” (4 points) [27]. Originally, the sum of the 11 items was multiplied by 10/11 after changing from “extremely rare” (0 points) to “most of the time” (3 points). However, since this study is a simple comparison of depression, only the total score was used. The higher the total, the higher the depression, and Cronbach’s α value was 0.68.

### 2.3. Analysis Methods

To minimize bias in the study, we used research tools with proven reliability and validity. We ensured anonymity using an online survey and applied consistent data collection procedures. To ensure an adequate sample size, increase the study’s power, and reduce the likelihood of random error affecting the results, we calculated the sample size using G*Power. The data for this study were analyzed using IBM SPSS 26.0. Frequency analysis was conducted to examine the general characteristics of the participants. To determine the differences between these characteristics and variables such as aging anxiety, menopausal quality of life, life satisfaction, and depression, T-tests, ANOVA, and Scheffé post hoc tests were utilized. Pearson correlation coefficients were calculated to assess the links among aging anxiety, menopausal quality of life, life satisfaction, and depression. We used both univariate and multivariate regression analyses to determine the factors associated with aging anxiety. First, we performed univariate analyses to identify significant predictors. Then we included these significant predictors (*p* < 0.05) in the multivariate regression model. Linear regression was employed to examine continuous outcomes. Mediation analysis was used to examine whether the relationship between the independent variable (menopausal quality of life) and the dependent variable (aging anxiety) is transmitted through the mediator variable (depression or life satisfaction). Unlike simple regression, which shows direct effects, mediation analysis reveals indirect pathways and the total effect of the independent variable on the dependent variable. The mediating effects of life satisfaction and depression on the relationship between menopausal quality of life and aging anxiety were analyzed using Process Macro Model 4. We included both depressive symptoms and physical health in the model to examine their combined effect on quality of life, allowing us to account for potential interdependence and interactions between these mediators.

## 3. Results

### 3.1. Differences in Variables According to Participants’ Characteristics

The age distribution of the participants was similar, with the highest average age group being 50–54 years, comprising 269 individuals (27.1%).There were 727 (73.2%) college graduates, outnumbering those with less than high school education, and 554 (55.8%) were employed. A total of 764 (76.9%) had a spouse, 441 (44.4%) had an average monthly household income of more than KRW 5 million, 555 (55.9%) had consumed alcohol in the past month, and 887 (89.3%) were nonsmokers. Regarding activity level and health, 615 (61.9%) reported exercising, 729 (73.4%) had a normal BMI (18.5–24.9), and 528 (53.2%) had self-reported good health. In terms of medical conditions, 356 (35.9%) had metabolic diseases, 59 (5.9%) had depression, 234 (23.6%) had joint diseases, and 121 (12.2%) had ophthalmic diseases (Table 1).

The outcomes of aging anxiety, the menopausal quality of life scale score, life satisfaction, and depression, stratified by general characteristics, are presented in Table 1. Aging anxiety was significantly linked to smoking (t = 3.28, *p* = 0.001), exercise (t = 2.05, *p* = 0.04), BMI (F = 8.99, *p* < 0.001), subjective health status (F = 22.44, *p* < 0.001), and depression (t = 4.59, *p* < 0.001).

The menopausal quality of life scale score was closely associated with education (F = 3.58, *p* = 0.028), mean monthly household income (F = 6.67, *p* = 0.001), alcohol consumption (t = 2.31, *p* = 0.021), smoking (t = 2.81, *p* = 0.006), exercise (t = 5.54, *p* < 0.001), subjective health status (F = 84.91 *p* < 0.001), metabolic diseases (t = 3.31, *p* = 0.001), depression (t = 6.10, *p* < 0.001), joint diseases (t = 6.53, *p* < 0.001), and ophthalmic diseases (t = 4.74, *p* < 0.001).

Life satisfaction demonstrated significant correlation with education (F = 10.03, *p* < 0.001), marital status (F = 23.06, *p* < 0.001), average monthly income (F = 50.73, *p* < 0.001), smoking (t = −4.82, *p* < 0.001), exercise (t = −6.91, *p* < 0.001), BMI (F = 3.55, *p* = 0.029), subjective health status (F = 60.63, *p* < 0.001), depression (t = −5.06, *p* < 0.001), and joint disease (t = −2.69, *p* = 0.007).

Depression was significantly associated with education (F = 4.52, *p* = 0.011), marital status (F = 11.90, *p* < 0.001), average monthly income (F = 13.04, *p* < 0.001), smoking (t = 5.55, *p* < 0.001), exercise (t = 6.48, *p* < 0.001), BMI (F = 3.79, *p* = 0.023), subjective health status (F = 59.56, *p* < 0.001), depression (t = 9.68, *p* < 0.001), joint disease (t = 3.27, *p* < 0.001), and ophthalmic disease (t = 2.68, *p* = 0.008).

### 3.2. Correlates with Aging Anxiety, Menopausal Quality of Life, Life Satisfaction, and Depression in Participants

Aging anxiety correlated positively with the menopausal quality of life scale score (r= 0.37, *p* < 0.001) and depression (r = 0.42, *p* < 0.001), and correlated negatively with life satisfaction (r= −0.46, *p* < 0.001). This indicates that, as the menopausal quality of life scale score and depression increase, aging anxiety also increases, whereas higher life satisfaction corresponds with lower aging anxiety among middle-aged women (Table 2).

### 3.3. Factors Affecting Aging Anxiety

Linear regression analysis was conducted using correlations and significant variables as input methods to identify the factors associated with aging anxiety (AAS) (Table 3). Independent variables included in the model were those identified as significant in univariate analyses. Prior to performing the analysis, multicollinearity among the independent variables was assessed. This revealed a variance inflation factor (VIF) of less than 10, indicating the absence of multicollinearity. The independence of errors was verified through the Durbin-Watson statistic, which yielded a value of 1.977, close to 2, confirming the lack of autocorrelation. Regression analysis demonstrated an adjusted R-squared value of 0.274, indicating that approximately 27.4% of the variance in aging anxiety is explained by the model. Significant predictors identified through the regression coefficient significance test included life satisfaction (t = −9.627, *p* < 0.001), depression CES-D (t = 4.879, *p* < 0.001), menopausal quality of life (t = 3.833, *p* < 0.001), and BMI (t = −3.851, *p* < 0.001). The standardization coefficient β represents the relative association of each independent variable with aging anxiety, which is the dependent variable, and life satisfaction had the greatest association (β = −0.310), followed by depression CES-D (β = 0.177), followed by the menopausal quality of life score (β = 0.134).

### 3.4. The Intermediary Association of Life Satisfaction and Depression with the Correlation between Menopausal Quality of Life and Aging Anxiety

To examine the mediating effects of depression and life satisfaction on the association between menopausal quality of life and aging anxiety, an analysis was conducted using SPSS’s Process Macro Model 4 (Table 4). The results from the significance tests of each pathway indicated that menopausal quality of life was significantly associated with depression (B = 1.483, t = 22.41, *p* < 0.001) and aging anxiety (B = 0.057, t = 4.352, *p* < 0.001). Additionally, depression had a significant association with aging anxiety (B = 0.028, t = 5.249, *p* < 0.001). Menopausal quality of life was significantly associated with life satisfaction (B = −0.277, t = −14.908, *p* < 0.001), and life satisfaction was significantly associated with aging anxiety (B = −0.181, t = −9.579, *p* < 0.001) (Table 4 and Figure 1). Menopausal quality of life, an independent variable, was thus significantly associated with the dependent variable, aging anxiety. Additionally, the mediating variables, depression and life satisfaction, exhibited a partial mediating effect.

### 3.5. Direct and Indirect Effects

Bootstrapping was employed to assess the significance of the indirect effects of depression and life satisfaction on the relationship between the menopausal quality of life score and aging anxiety [28]. The partial mediation of the path from the menopausal quality of life score to aging anxiety via depression and life satisfaction was bootstrapped 5000 times. The results indicated that the pathway from the menopausal quality of life score to aging anxiety through depression and life satisfaction was significant, as the 95% confidence interval of the indirect effect did not include zero. Consequently, life satisfaction and depression exerted an indirect effect on the relationship between the menopausal quality of life score and aging anxiety (Table 5).

## 4. Discussion

Menopausal symptoms may reduce life satisfaction and heighten the risk of depression in middle-aged women, thereby exacerbating aging anxiety. This study aimed to identify the association between menopausal quality of life and aging anxiety. Additionally, it sought to examine the mediating effects of life satisfaction and depression on the relationship between menopausal quality of life and aging anxiety.

The mean aging anxiety score of middle-aged women in this study was 3.20 out of 5. This was slightly higher than the mean aging anxiety score of 3.00 in a study of aging anxiety in late middle-aged women using the same scale [29].The score was similar to the mean score of 3.24 in a study by Chang [30] using an aging anxiety scale developed for middle-aged women [31]. Smoking, exercise, BMI, subjective health status, and depression were factors of aging anxiety among the participants. Aging anxiety was the highest in underweight participants with a BMI of less than 18.5, which was different from the results Lim’s study [32] which found the highest aging anxiety in obese participants. However, this difference was not significant in Lim’s study. Due to the limited research on the relationship between BMI and aging anxiety, it is challenging to draw definitive conclusions. However, considering the established relationship between BMI and depression, it can be inferred that similar patterns may exist. Studies have shown that individuals who are underweight are at a higher risk of depression. For instance, Hong’s study reported that 11.3% of the underweight group experienced depression, compared to 8.3% in the second-stage obesity group and 6.2% in the normal weight group, indicating that the underweight group is more susceptible to depression [33]. These findings are consistent with the current study’s results, which revealed the highest levels of depression in individuals with varying BMI categories.

MENQOL is a globally recognized tool for assessing menopausal quality of life. This study utilized the Korean version of the MENQOL, translated and validated by Park et al., to ensure reliability and validity. The menopausal quality of life, in relation to the general characteristics of the participants, showed statistically significant differences based on education level, average monthly household income, alcohol consumption, smoking, exercise habits, subjective health status, and presence of depression, metabolic, joint, and ocular diseases. These findings align with a previous study [34], which indicated that menopausal women are at increased risk for cardiovascular disease due to decreased estrogen levels, reduced HDL cholesterol, decreased bone mass leading to osteoporosis, and heightened depression and stress stemming from negative emotions.

Although this study exclusively assessed self-reported menopausal status, other studies have utilized physiological markers such as an increase in follicle-stimulating hormone (FSH) or a decrease in estradiol (E2) in blood samples to confirm menopausal status. Recently, a comparative study investigated the quality of life and attitudes toward aging among women undergoing hormone replacement therapy (HRT) for alleviating mental and psychological symptoms associated with menopause, contrasting them with those who did not receive HRT. Results indicated that the HRT group exhibited more favorable attitudes toward both menopause and aging, enhanced social network functioning, and a higher overall quality of life [35]. Additionally, another study measured FSH and E2 levels and correlated vasomotor symptoms (VMS) and MENQOL with menopausal status and depressive symptoms. The findings revealed that menopausal status, depressive symptoms, and poor health were significantly associated with all four domains of MENQOL [36].

SWLS, developed by Diener et al. [25], is a measure of overall life satisfaction. In this study, significant differences were observed in life satisfaction based on education, marital status, monthly household income, smoking, exercise, body mass index, subjective health status, depression, and joint disease. These findings are consistent with those of previous studies that demonstrated higher life satisfaction among individuals who are married, possess higher levels of education and economic status, and enjoy good health [37].

CES-D was originally designed by Radloff [38] as a 20-item questionnaire, but the CESD-11 scale [26] was used in this study to simply compare levels of depression and not to determine depression. Eo [39] investigated the complex interactions between depression and various sociodemographic factors in Korean menopausal women, reporting significant influences from factors such as age, education, income, employment status, and marital status. Park [40] emphasized that participation in health promotion activities such as regular physical exercise, balanced nutrition, and stress management can greatly mediate the negative effects of menopausal symptoms of depression.

The study examined the interplay between menopausal quality of life, life satisfaction, depression, and aging anxiety in middle-aged women. It found that life satisfaction and depression significantly mediated the relationship between menopausal quality of life and aging anxiety. Additionally, other research demonstrated that physical symptoms associated with menopause, such as sleep disturbances and stress, diminish overall quality of life. For instance, Amira’s study revealed a correlation between more severe menopausal symptoms and heightened psychological distress among middle-aged Emirati women [41]. Similarly, Anula’s study underscored that menopausal symptoms contribute to anxiety and depression, which subsequently impact sleep quality [42].

Recognizing these factors is crucial, and developing strategies to address them is imperative. Enhancing menopausal quality of life represents a foundational approach to mitigating depression and enhancing life satisfaction, thereby potentially reducing aging anxiety.

This study was limited to women aged 45–65 years residing in Seoul, Korea, which may affect the generalizability of the findings. Additionally, the use of a recall-based self-report questionnaire could introduce recall bias. A major limitation of this study is its cross-sectional design, which precludes causal inferences. Longitudinal studies are needed to confirm the directionality of the relationships observed. Another major limitation is that the MENQOL does not include specific physiological markers or measurements to objectively quantify menopause-related changes, focusing instead on subjective assessments of quality of life related to menopausal symptoms. Future studies should aim to provide additional validation data on the MENQOL, such as correlations with known physiological markers from other research. Despite these limitations, the study is notable for examining the partial mediating effects of life satisfaction and depression on aging anxiety using a menopausal-specific quality of life scale tailored to Korean women.

## 5. Conclusions

To effectively address and reduce the anxiety related to aging, as highlighted by this study, mental health programs specifically tailored for menopausal women, must be developed. These programs should focus on the distinct psychological needs of these women, offering targeted support and resources to help manage their anxiety.

Furthermore, comparative studies must be conducted between Korean women and women from other countries. These studies will provide valuable insights into the differences and similarities in aging anxiety across various cultures and societies, enabling the development of universally applicable or culturally specific strategies.

Additionally, a longitudinal study is recommended to thoroughly understand the causal relationships and long-term effects of the factors identified in this study. By observing changes over an extended period, we can gain a deeper understanding of how these factors are associated with aging anxiety over time.

Implementing these recommendations will enable us to create more effective interventions and support systems for menopausal women, ultimately enhancing their mental health and overall well-being.

## Figures and Tables

**Figure 1 medicina-60-01189-f001:**
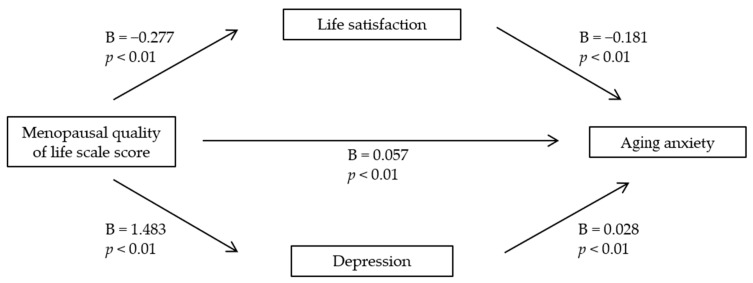
Mediating effects of life satisfaction and depression on the association between the menopausal quality of life scale score and aging anxiety.

**Table 1 medicina-60-01189-t001:** Difference in AAS, MENQOL-K, SWLS, and CES-D according to the Participant Characteristics (N = 993).

Variables	Categories	N (%)	AAS		MENQOL-K		SWLS		CES-D	
			Mean ± SD	F or T (*p*)	Mean ± SD	F or T (*p*)	Mean ± SD	F or T (*p*)	Mean ± SD	F or T (*p*)
Age (year)	45–49	242 (24.4)	3.24 ± 0.55	1.20 (0.309)	3.18 ± 1.34	1.38 (0.248)	2.81 ± 0.82	0.08 (0.971)	10.83 ± 3.56	2.06 (0.104)
	50–54	269 (27.1)	3.21 ± 0.50		3.33 ± 1.30		2.81 ± 0.84		10.15 ± 3.38	
	55–59	235 (23.7)	3.20 ± 0.48		3.41 ± 1.28		2.78 ± 0.88		10.56 ± 3.21	
	60–64	247 (24.9)	3.15 ± 0.49		3.34 ± 1.28		2.82 ± 0.83		10.31 ± 3.10	
Education	≤Middle school	11 (1.1)	3.21 ± 0.45	0.00 (0.998)	3.58 ± 1.68	3.58 (0.028) *	2.71 ± 1.22	10.03 (<0.001) **	11.05 ± 3.78	4.52 (0.011) *
	High School	255 (25.7)	3.20 ± 0.46		3.49 ± 1.42		2.61 ± 0.87		10.97 ± 3.41	
	≥college	727 (73.2)	3.20 ± 0.52		3.25 ± 1.24		2.88 ± 0.81		10.26 ± 3.27	
Occupation	Employed	554 (55.8)	3.20 ± 0.50	0.11 (0.910)	3.28 ± 1.32	−0.87 (0.385)	2.77 ± 0.85	−1.55 (0.121)	10.55 ± 3.28	1.05 (0.295)
	Unemployed	439 (44.2)	3.20 ± 0.50		3.36 ± 1.27		2.85 ± 0.83		10.33 ± 3.38	
Marital status	Married	764 (76.9)	3.18 ± 0.49	2.03 (0.132)	3.30 ± 1.26	1.71 (0.182)	2.90 ± 0.81	23.06 (<0.001) **	10.18 ± 3.09	11.90 (<0.001) **
	Divorced/Widowed	142 (14.3)	3.28 ± 0.54		3.49 ± 1.47		2.40 ± 0.85		11.53 ± 4.02	
	Single	87 (8.8)	3.22 ± 0.58		3.20 ± 1.34		2.66 ± 0.87		11.10 ± 3.66	
Income (unit: 10,000 won)	<300	242 (24.4)	3.24 ± 0.50	1.32 (0.267)	3.51 ± 1.43	6.67 (0.001) *	2.41 ± 0.83	50.73 (<0.001) **	11.16 ± 3.58	13.04 (<0.001) **
	300–500	310 (31.2)	3.21 ± 0.49		3.39 ± 1.27		2.76 ± 0.82		10.71 ± 3.41	
	>500	441 (44.4)	3.17 ± 0.51		3.16 ± 1.22		3.05 ± 0.78		9.89 ± 3.02	
Drinking alcohol	Yes	555 (55.9)	3.23 ± 0.50	1.82 (0.069)	3.40 ± 1.31	2.31 (0.021) *	2.80 ± 0.81	−0.19 (0.853)	10.60 ± 3.28	1.55 (0.122)
	No	438 (44.1)	3.17 ± 0.50		3.21 ± 1.28		2.81 ± 0.88		10.27 ± 3.37	
Smoking	Yes	106 (10.7)	3.37 ± 0.57	3.28 (0.001) *	3.69 ± 1.48	2.81 (0.006) *	2.39 ± 0.95	−4.82 (<0.001) **	12.48 ± 4.08	5.55 (<0.001) **
	No	887 (89.3)	3.18 ± 0.49		3.27 ± 1.27		2.86 ± 0.82		10.21 ± 3.14	
Exercise	Yes	615 (61.9)	3.17 ± 0.50	2.05 (0.040) *	3.14 ± 1.26	5.44 (<0.001) **	2.95 ± 0.82	−6.91 (<0.001) **	9.93 ± 3.01	6.48 (<0.001) **
	No	378 (38.1)	3.24 ± 0.50		3.60 ± 1.32		2.58 ± 0.83		11.31 ± 3.63	
BMI	<18.5	70 (7.0)	3.34 ± 0.52	8.99 (<0.001) **	3.28 ± 1.37	0.14 (0.867)	2.55 ± 0.84	3.55 (0.029) *	11.43 ± 4.37	3.79 (0.023) *
	18.5 to <25	729 (73.4)	3.22 ± 0.50		3.31 ± 1.31		2.83 ± 0.86		10.43 ± 3.28	
	≥25	194 (19.5)	3.08 ± 0.49		3.36 ± 1.24		2.82 ± 0.77		10.17 ± 3.00	
Subjective health status	Unhealthy	216 (21.8)	3.38 ± 0.55	22.44 (<0.001) **	4.10 ± 1.28	84.91 (<0.001) **	2.37 ± 0.82	60.63 (<0.001) **	12.24 ± 3.82	59.56 (<0.001) **
	Moderate	528 (53.2)	3.19 ± 0.44		3.31 ± 1.23		2.81 ± 0.74		10.38 ± 3.03	
	Healthy	249 (25.1)	3.08 ± 0.54		2.64 ± 1.06		3.18 ± 0.89		9.05 ± 2.70	
Metabolic disease	Yes	356 (35.9)	3.20 ± 0.51	0.14 (0.889)	3.21 ± 1.30	3.31 (0.001) *	2.84 ± 0.84	−1.71 (0.089)	10.35 ± 3.33	1.34 (0.181)
	No	637 (64.1)	3.20 ± 0.49		3.50 ± 1.29		2.74 ± 0.85		10.64 ± 3.31	
Depression	Yes	59 (5.9)	3.18 ± 0.50	4.59 (<0.001) **	3.25 ± 1.28	6.10 (<0.001) **	2.84 ± 0.83	−5.06 (<0.001) **	10.21 ± 3.14	9.68 (<0.001) **
	No	934 (94.1)	3.49 ± 0.54		4.30 ± 1.28		2.27 ± 0.82		14.34 ± 3.70	
Joint disease	Yes	234 (23.6)	3.19 ± 0.51	1.57 (0.117)	3.17 ± 1.27	6.53 (<0.001) **	2.85 ± 0.83	−2.69 (0.007) *	10.25 ± 3.18	3.27 (<0.001) **
	No	759 (76.4)	3.25 ± 0.49		3.79 ± 1.30		2.68 ± 0.86		11.12 ± 3.68	
Ophthalmic disease	Yes	121 (12.2)	3.19 ± 0.50	1.44 (0.149)	3.24 ± 1.29	4.74 (<0.001) **	2.82 ± 0.84	−1.88 (0.060)	10.35 ± 3.28	2.68 (0.008) *
	No	872 (87.8)	3.26 ± 0.50		3.83 ± 1.27		2.67 ± 0.81		11.21 ± 3.55	

* *p* < 0.05, ** *p* < 0.01. SD = Standard deviation; AAS = Aging anxiety scale; MENQOL-K = Korean version of menopause-specific quality of life assessment; SWLS = Satisfaction with life scale; CES-D = Center for the epidemiologic studies—depression scale.

**Table 2 medicina-60-01189-t002:** Correlations among study variables (N = 993).

Variables	AAS	MENQOL-K	SWLS	CES-D
AAS	1			
MENQOL-K	0.37 (<0.001)	1		
SWLS	−0.46 (<0.001)	−0.43 (<0.001)	1	
CES-D	0.42 (<0.001)	0.58 (<0.001)	−0.49 (<0.001)	1

AAS = Aging anxiety scale; MENQOL-K = Korean version of menopause-specific quality of life assessment; SWLS = Satisfaction with life scale; CES-D = Center for the epidemiologic studies—depression scale.

**Table 3 medicina-60-01189-t003:** Factors affecting aging anxiety (N = 993).

Variables	B	SE	ß	T	*p*	Adj R^2^	F(*p*)
Constant	61.854	0.1.817		34.049	<0.001	0.274	42.588 (<0.001)
SWLS	−0.675	0.070	−0.310	−9.627	<0.001		
CES-D	0.269	0.055	0.177	4.879	<0.001		
MENQOL-K	0.959	0.250	0.134	3.833	<0.001		
Smoking	0.145	0.833	0.005	0.174	0.862		
BMI	−2.458	0.638	−0.106	−3.851	<0.001		
Subjective health status	−0.374	0.667	−0.020	−0.560	0.576		
Depression	0.195	1.109	0.005	0.176	0.860		
Ophthalmic disease	0.169	0.618	0.008	0.273	0.785		

MENQOL-K = Korean version of menopause-specific quality of life assessment; SWLS = Satisfaction with life scale; CES-D = Center for the epidemiologic studies—depression scale; SE = Standard error.

**Table 4 medicina-60-01189-t004:** Mediating effects of SWLS and CES-D in the relationship between MENQOL-K and AAS (N = 993).

Variables			B	SE	T	*p*	LLCI	ULCI
MENQOL-K	→	CES-D	1.483	0.066	22.41 **	<0.001	1.353	1.613
MENQOL-K	→	AAS	0.057	0.013	4.352 **	<0.001	0.031	0.083
MENQOL-K	→	SWLS	−0.277	0.019	−14.908 **	<0.001	−0.314	−0.241
CES-D	→	AAS	0.028	0.005	5.249 **	<0.001	0.017	0.038
SWLS	→	AAS	−0.181	0.019	−9.579 **	<0.001	−0.219	−0.144

AAS = Aging anxiety scale; MENQOL-K = Korean version of menopause-specific quality of life assessment; SWLS = Satisfaction with life scale; CES-D = Center for the epidemiologic studies—depression scale; SE = Standard error; LLCI = lower limit confidence interval; ULCI = upper limit confidence interval; → = Path; ** = *p* < 0.01.

**Table 5 medicina-60-01189-t005:** Direct and indirect effects on AAS (N = 993).

Variable			Effect	BootSE	BootLLCI	BootULCI
MENQOL-K		AAS	0.057	0.013	0.031	0.083
MENQOL-K	CES-D	AAS	0.041	0.008	0.025	0.058
MENQOL-K	SWLS	AAS	0.050	0.006	0.038	0.064
Total			0.092	0.01	0.073	0.111

AAS = Aging anxiety scale; MENQOL-K = Korean version of menopause-specific quality of life assessment; SWLS = Satisfaction with life scale; CES-D = Center for the epidemiologic studies—depression scale; SE = Standard error; LLCI = lower limit confidence interval; ULCI = upper limit confidence interval.

## Data Availability

The datasets used and/or analyzed during the current study are restricted due to their sensitivity but are available from the corresponding authors upon reasonable request.

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
