# Peer review of "Association between Menopausal Women’s Quality of Life and Aging Anxiety: The Role of Life Satisfaction and Depression"

_medicina, 2024, doi:10.3390/medicina60081189_

Round 1

Reviewer 1 Report

Comments and Suggestions for Authors

The submitted manuscript entitled "he influence of quality of life on aging anxiety in postmenopausal women: the link between life satisfaction and depression" describes a single study on the associations between menopausal quality of life, life satisfaction, depression, and aging anxiety. The strengths of the study are large sample and validated measurements. However, before suggesting it for publication in Medicina, the below pointed issues should be addressed.

#1. The design of the study was cross-sectional. Any causal interpretation (e.g., using a word "influence") are not possible. The Authors examined only the associations between the variables. Thus, starting from the title, the manuscript should be screened and corrected for phrases suggesting causality.

#2. The introduction discussed main variables. However, the mediating mechanisms was not justified properly. Particularly in cross-sectional design, the justification of the hypotheses concerning the mediating mechanisms should be elaborated.

#3. In the introduction there was no direct reference to such variables as BMI, which were measured later in the study. In my opinion, if the Authors planned to analyse the role of such variables, they should introduce them in the introduction part of the manuscript.

#4. I was a little confused by the menopausal quality of life measurement. The higher score indicated low quality of life based on menopausal symptoms. I am wondering whether the Authors could reverse the score so the higher score indicate higher quality of life despite menopasal symptoms.

#5. In my opinion, the core drawback of the study was lack of any variable measuring the physiological index of menopause. GIven the wide range of the age, we don't really know whether women participating in the study were in pre or postmenopausal stage. Alternatively, the Auhtors could provide more data on the validity of the measurement of menopausal quality of life (e.g., some correlations with objective measures of physiological aging).

#6. Did the Authors introduce both mediatiors in the same model? Did they compare the mediating chains?

#7. The discussion section is very basic. The Authors simply summarize the results. In my opinion, they should be discussed more in-depth. How lower quality of life because of menopausal symptoms is related with higher depression. Which are the reasons of this association and potential moderating factors (e.g., dyadic coping).

Comments on the Quality of English Language

The manuscript would benefit from the professional language editing.

Author Response

Reviewer 1.

#1. The design of the study was cross-sectional. Any causal interpretation (e.g., using a word "influence") are not possible. The Authors examined only the associations between the variables. Thus, starting from the title, the manuscript should be screened and corrected for phrases suggesting causality.

Response: Thank you for your pertinent comment. We have corrected all the words and phrases that imply causality and replaced them with words that emphasize that it is the associations that were studied. See the title on page 1.

#2. The introduction discussed main variables. However, the mediating mechanisms was not justified properly. Particularly in cross-sectional design, the justification of the hypotheses concerning the mediating mechanisms should be elaborated.

Response: Thank you for highlighting this. Following your recommendation, we have added a description of mediating mechanisms in the introduction and added references to the literature (pp. 2, lines 90-101).

#3. In the introduction there was no direct reference to such variables as BMI, which were measured later in the study. In my opinion, if the Authors planned to analyse the role of such variables, they should introduce them in the introduction part of the manuscript.

Response: We apologize for this discrepancy. To address this, we have added a description of BMI in the Introduction and the relevant literature on its impact on women’s quality of life (p. 2, lines 82-89).

#4. I was a little confused by the menopausal quality of life measurement. The higher score indicated low quality of life based on menopausal symptoms. I am wondering whether the Authors could reverse the score so the higher score indicate higher quality of life despite menopasal symptoms.

Response: Thank you for highlighting this. We realize the confusion this may cause. Accordingly, we have modified the explanation of the menopausal quality of life scoring system to make it more intuitive and easier to understand: the higher the score on the menopausal quality of life scale, the more severe the symptoms and thus the lower the quality of life. Although we considered reversing the scores for clarity, we decided to keep the original scoring system for consistency with previous studies (p. 3, lines 132-145).

#5. In my opinion, the core drawback of the study was lack of any variable measuring the physiological index of menopause. GIven the wide range of the age, we don't really know whether women participating in the study were in pre or postmenopausal stage. Alternatively, the Auhtors could provide more data on the validity of the measurement of menopausal quality of life (e.g., some correlations with objective measures of physiological aging).

Response: We acknowledge your point. Accordingly, we have introduced data from similar studies that include physiological indices and think it should be studied further in future studies. Nonetheless, we have acknowledged this as a limitation of this study in the discussion (p.10, line 324-336).

#6. Did the Authors introduce both mediatiors in the same model? Did they compare the mediating chains?

Response: Yes. In the Methods, we explained that both mediators were introduced in the same mediation model (p. 4, lines 175-184).

We described the mediation effects analysis in the Results (p. 8, lines 255-266).

#7. The discussion section is very basic. The Authors simply summarize the results. In my opinion, they should be discussed more in-depth. How lower quality of life because of menopausal symptoms is related with higher depression. Which are the reasons of this association and potential moderating factors (e.g., dyadic coping).

Response: Thank you for this constructive comment. We have expanded our discussion of the association of menopausal symptoms with quality of life, depression, and anxiety by comparing our results with those reported in the literature (p. 10, lines 352-364).

Reviewer 2 Report

Comments and Suggestions for Authors

 The study entitled:The influence of quality of life on aging anxiety in postmeno-2 pausal women: the link between life satisfaction and depression” address an important topic however, there are several concerns to be addressed

The title and the abstract background are quite complicated and not understood

The authors have to redraft them in a simple direct and comprehendible way

The study sample is fine, however, more information is required to support the minimization of the bias especially for a self-administered study tool

The authors has to answer the question: what is new in this study? The association between depression, life satisfaction, anxiety is already established, why to conduct this study?

The ‘Subsection” should be removed from the methods

The data analysis reads: “Factors influencing aging anxiety were examined through multiple 146 regression analysis”, the reader likes more details, i.e was this multivariable regression preceded by an univariate ? was it a backward model? Linear or logistic?

The results subheading is complex , for example:” 3.1. Aging anxiety, menopausal quality of life scale, life satisfaction, and depression by general  characteristics” should be simplified

In Table 3, what was the statistical analysis done?

Was this a linear regression?

Please explain what is the meaning of “Mediating effect” to the reader, how does it differ from regression? Why it was used?

In figure 1, is this a mediating effect? Or person relation?

Table 5. Direct and indirect effects on AAS (N=993). What is the direct and indirect effect? Part of what? They were not mentioned in the data analysis? How they are calculated and what is their rationale?

The cross-sectional design is another limitation  

Author Response

Reviewer 2.

#1. The title and the abstract background are quite complicated and not understood. The authors have to redraft them in a simple direct and comprehendible way

Response: Thank you for pointing this out. We have simplified both the title and abstract background to be more direct and easier to comprehend. (pp. 1-title and abstract).

#2. The study sample is fine, however, more information is required to support the minimization of the bias especially for a self-administered study tool

Response: Thank you for this pertinent comment. We have explained how we minimized bias in the Methods (pp. 4, lines 161-165).

#3. The authors has to answer the question: what is new in this study? The association between depression, life satisfaction, anxiety is already established, why to conduct this study?

Response: While previous studies have established associations between depression, life satisfaction, and anxiety, this study is unique in that it examined these relationships in an under-studied population, Korean menopausal women, taking into account cultural and demographic characteristics. In addition, it examined the mediating roles of life satisfaction and depression in these associations, providing a more nuanced understanding (pp. 2, line 102-108).

#4. The ‘Subsection” should be removed from the methods

Response: We have replaced “subsection” with “Participants” (pp. 3, line 111).

#5. The data analysis reads: “Factors influencing aging anxiety were examined through multiple 146 regression analysis”, the reader likes more details, i.e was this multivariable regression preceded by an univariate ? was it a backward model? Linear or logistic?

Response: Thank you for your attention to detail. We have added the requisite description of the statistical analyses in the Methods (pp. 4, lines 171-175).

#6. The results subheading is complex , for example:” 3.1. Aging anxiety, menopausal quality of life scale, life satisfaction, and depression by general  characteristics” should be simplified

Response: We apologize for the cumbersome subheading. We have simplified it accordingly (pp. 4, line 186).

#7. In Table 3, what was the statistical analysis done?

Was this a linear regression?

Response: Yes, this was a linear regression. This has been described better under “3.3. Factors Affecting Aging Anxiety” (pp. 7, line 231-235).

#8. Please explain what is the meaning of “Mediating effect” to the reader, how does it differ from regression? Why it was used?

Response: Thank you for these clarity seeking questions. In response, we have explained the meaning of “mediating effects” for the benefit of the reader (pp. 2, lines 90-101).

#9. In figure 1, is this a mediating effect? Or person relation?

Response: It is a mediating effect as you surmised (pp. 9, lines 274-275).

#10. Table 5. Direct and indirect effects on AAS (N=993). What is the direct and indirect effect? Part of what? They were not mentioned in the data analysis? How they are calculated and what is their rationale?

Response: These effects are explained in the Introduction and in Section 3.5 Direct and Indirect Effects (pp. 2, lines 99-101; pp. 9, lines 277-280).

#11. The cross-sectional design is another limitation

Response: We recognize the limitation that you point out and have added it to the Discussion. Additionally, we noted that longitudinal studies are needed to confirm the directionality of the observed relationships (pp. 11, lines 367-374).

Round 2

Reviewer 2 Report

Comments and Suggestions for Authors

The manuscript has been improved 

The authors performed the changes required